# Cardioprotective Effects of Grapefruit IntegroPectin Extracted via Hydrodynamic Cavitation from By-Products of *Citrus* Fruits Industry: Role of Mitochondrial Potassium Channels

**DOI:** 10.3390/foods11182799

**Published:** 2022-09-11

**Authors:** Lorenzo Flori, Lorenzo Albanese, Vincenzo Calderone, Francesco Meneguzzo, Mario Pagliaro, Rosaria Ciriminna, Federica Zabini, Lara Testai

**Affiliations:** 1Department of Pharmacy, University of Pisa, 56126 Pisa, Italy; 2Istituto per la Bioeconomia, CNR, Via Madonna del Piano 10, 50019 Sesto Fiorentino, Italy; 3Interdepartmental Center of Nutrafood, University of Pisa, Via Del Borghetto, 56120 Pisa, Italy; 4Istituto per lo Studio dei Materiali Nanostrutturati, CNR, Via U. La Malfa 153, 90146 Palermo, Italy

**Keywords:** *Citrus* flavonoids, naringin, naringenin, pectin, byproducts, anti-ischemic myocardial protection, hydrodynamic cavitation

## Abstract

*Citrus* flavonoids are well-known for their beneficial effects at the cardiovascular and cardio-metabolic level, but often the encouraging in vitro results are not confirmed by in vivo approaches; in addition, the clinical trials are also inconsistent. Their limited bioavailability can be, at least in part, the reason for these discrepancies. Therefore, many efforts have been made towards the improvement of their bioavailability. Hydrodynamic cavitation methods were successfully applied to the extraction of byproducts of the *Citrus* fruits industry, showing high process yields and affording stable phytocomplexes, known as IntegroPectin, endowed with great amounts of bioactive compounds and high water solubility. The cardioprotective effects of grapefruit IntegroPectin were evaluated by an ex vivo ischemia/reperfusion protocol. Further pharmacological characterization was carried out to assess the involvement of mitochondrial potassium channels. Grapefruit IntegroPectin, where naringin represented 98% of the flavonoids, showed anti-ischemic cardioprotective activity, which was better than pure naringenin (the bioactive aglycone of naringin). On cardiac-isolated mitochondria, this extract confirmed that naringenin/naringin were involved in the activation of mitochondrial potassium channels. The hydrodynamic cavitation-based extraction confirmed a valuable opportunity for the exploitation of *Citrus* fruits waste, with the end product presenting high levels of *Citrus* flavonoids and improved bioaccessibility that enhances its nutraceutical and economic value.

## 1. Introduction

*Citrus* flavonoids are well-known for their beneficial effects at the cardiovascular and cardio-metabolic level [1]. Recently, daily hesperidin supplementation was shown to improve the blood pressure in pre- and stage 1 phase hypertensive patients [2]. Likewise, the *Citrus* flavonoids naringin and naringenin, the latter being the aglycone of naringin, are promising nutraceuticals in the strategy regarding the management of cardiovascular complications, improving the systolic pressure levels and the metabolic profile [3]. In vitro and in vivo evidence suggests that multiple pathways may be involved in these effects, including the positive modulation of the sirtuin 1 (SIRT1) enzyme pathway [4,5,6]. However, other putative mechanisms have been recognized in these biological actions, such as the stimulation of potassium channels, located both on sarcolemmal and on inner membranes of cardiac mitochondria, and the increase in bioavailability of nitric oxide, contributing to the preservation of the endothelial barrier at the vascular level [7,8,9,10]. Of note, two of the most studied *Citrus* flavonoids, hesperetin and naringenin, were shown to be endowed with cardioprotective effects. For example, hesperetin protects the heart against the toxicity induced by doxorubicin treatment [11], while naringenin exerts protection against ischemia/reperfusion injury, both in young-adult and in 12-month aged rats, through the activation of mitochondrial calcium-activated potassium channels (mitoBK) [7,12]. Recently, naringin was also reported to be an activator of BK channels [13,14].

Despite the fact that numerous pre-clinical studies suggest *Citrus* flavonoids’ nutraceutical value in the maintenance of cardiovascular and metabolic homeostasis, often clinical trials show inconsistencies, probably due to the poor systemic bioavailability of *Citrus* flavonoids. Many efforts have been made in order to improve their pharmacokinetic profile, mainly attempting to make secondary metabolites more soluble and more accessible at the intestinal level, via structural transformation (i.e., glycosylation) and pharmacological technologies [15]. 

In the last few years, hydrodynamic cavitation (HC) technology has been successfully applied to the extraction of waste streams of the *Citrus* fruits industry [16]. This technique surprisingly leads to extracts rich in stable phyto-complexes comprising pectin, flavonoids and volatiles. These complexes are known as “IntegroPectin”, which can be extracted from the water phase by means of a standard drying technique, e.g., freeze-drying [17,18]. The innovative HC technique allows us to achieve high value-added products from waste streams and byproducts, including from *Citrus* fruits, complying with the principles of circular economy, as well as with the principles of green extraction of natural products [19]. 

Grapefruit IntegroPectin showed remarkable water solubility, which was directly related to the cavitation-based extraction process, as well as in vitro bactericidal activity against both Gram-negative and Gram-positive bacteria, that was much higher than commercial *Citrus* pectins [20]. Even more important for the purposes of this article, in a previous study carried out on a human neuroblastoma cells line, grapefruit IntegroPectin was shown to be devoid of cytotoxicity effect and, on cells exposed on oxidative stress, it preserved mitochondrial membrane potential and cell morphology and demonstrated powerful antiproliferative activity. Interestingly, similar commercial *Citrus* pectins showed lower protective efficacy, if compared to IntegroPectin. The authors hypothesized that terpenes and flavonoids may be adsorbed on the pectic surface rich in RG-I “hairy” regions, containing galactose and arabinose units, making the biologically active secondary metabolites more bio-accessible [21].

In this study, we investigated, for the first time, the in vivo efficacy of grapefruit IntegroPectin in a model of myocardial ischemia/reperfusion injury.

## 2. Materials and Methods

### 2.1. Production of Grapefruit IntegroPectin

The grapefruit IntegroPectin was isolated via freeze-drying of the water phase extract upon hydrodynamic cavitation of waste peels of organic pink grapefruit, kindly donated by OPAC Campisi (Siracusa, Italy). The details of the hydrodynamic cavitation-based extractor, comprising a closed hydraulic circuit with a centrifugal pump and a Venturi-shaped reactor, with electricity as the only energy source, were described in a previous study about the extraction of waste orange peel [16]. The details of the specific process for the extraction of waste grapefruit peels were as follows: fresh biomass in the amount of 34 kg, mixed with 120 L of water, with no other additives; process time of 1 h, carried out at atmospheric pressure; free heating from 7.5 °C to 38 °C; overall consumed energy at the level of 0.2 kWh per kg of fresh waste grapefruit peel. Isolated via filtering, centrifugation at 10^4^× *g* rpm in 80 mL Falcon probes and freeze drying of the resulting surnatant. The newly obtained grapefruit IntegroPectin is a low pectin rich in adsorbed naringin and hesperidin [22], as well as in highly bioactive terpenes α-terpineol, terpinen-4-ol, linalool and limonene [23].

Figure 1 shows a schematic diagram for the production of grapefruit IntegroPectin.

### 2.2. Animal Experimentation and Data Analysis

Male Wistar rats of about 12–15 weeks and weight between 300 and 400 g were used. The animals were housed in cages, with freedom of movement, supplied with water and food, and exposed to light/dark cycles of 12 h each, at 22 °C. The study was carried out in line with EU legislation (EEC Directive 63/2010) and Italian legislation (Legislative Decree No. 26/2014) on the protection of animals used for scientific purposes. Formal approval to conduct the described experiments was obtained from the Animal Subjects Review Board of the University of Pisa (protocol number 491/2018-PR). With regards to the ex vivo approach, we discussed it and obtained the following protocol number: 45975/2016.

#### 2.2.1. Ischemia/Reperfusion on Langendorff Isolated and Perfused Heart

The animals were treated with intraperitoneal injection (i.p.) of naringenin (100 mg/Kg), vehicle (dimethyl sulfoxide, DMSO, 1 mL/Kg) or grapefruit IntegroPectin (45 mg/kg; 135 mg/kg; 450 mg/kg), 2 h before heart removal. 

The animals were subjected to an i.p. injection of heparin (2500 I.U.) (Sigma-Aldrich, St. Louis, MO, USA), and to a subsequent one of pentothal Sodium (100 mg/kg) (MSD Italia) to induce a deep anesthesia. Each animal was sacrificed, and the explanted heart was immersed in a Krebs solution (NaHCO_3_ 25.0 mM, Glucose 11.7 mM, NaCl 117.9 mM, KCl 4.8 mM, MgSO_4_ 2.5 mM, CaCl_2_ 2.2 mM, KH_2_PO_4_ 1.2 mM; clioxicarb: 95% O_2_ and 5% CO_2_; pH 7.4; 4 °C). Then, the heart was set up on a Langendorff apparatus and perfused with Krebs saline solution at 37 °C and constant pressure of 70–80 mmHg, through a peristaltic pump (Peri-Star, 2Biological Instruments, Besozzo, Italy). A latex balloon filled with bidistilled water at a pressure of 5–10 mmHg was placed inside the left ventricle through the mitral valve to monitor the functional parameters of the heart. The ischemia/reperfusion protocol consisted of 30 min stabilization time, 30 min of global ischemia and 120 min of reperfusion. At the end of reperfusion time, the heart was removed, dried, weighed, then the left ventricle was isolated. This was sliced into cross sections and immersed in a 1% *w*/*v* solution of 2,3,5-triphenyltetrazolium chloride (TTC) (Sigma-Aldrich) dissolved in PBS (pH 7.4) (Sigma-Aldrich) for 20 min at 37 °C, in the dark. Finally, the slices were fixed in a 10% *v*/*v* aqueous formaldehyde solution. As a result, it gave the vital areas a red color. 

The latex balloon was connected to a pressure transducer (Bentley Trantec, mod. 800, Ugo Basile, Comerio, Italy) and a data acquisition system (Biopac Systems Inc., Goleta, CA, USA). Recorded parameters were left ventricular pressure (LVDP), heart rate (HR), and contraction/relaxation time (dp/dt). The rate-pressure product (RPP) was calculated as the product of the first two parameters. The effects of the different doses of grapefruit IntegroPectin on cardiac functional parameters (RPP, dp/dt) were expressed in % of the basal values measured at the end of the stabilization period.

The cardiac areas involved in ischemia/reperfusion damage, necrotic or apoptotic, remained white or pale pink. Ischemic area was calculated as % of left ventricle total area (Ai/Avs %) using the software GIMP (release 2.10.32, Spencer Kimball and Peter Mattis, Berkeley, CA, USA).

All data were obtained from 6 different animals and graphed using the GraphPad Prism 8.0 program and expressed as mean ± standard error of mean (SEM). Statistical analyzes were performed with the Student’s *t*-test, with *p* < 0.05 considered as an indicator of significant difference.

#### 2.2.2. Mitochondrial Isolation Protocol

The animals were sacrificed after isoflurane anesthesia (R584S Small Animal Aesthesia Machine, Shenzhen RWD Life Science and Technology Co., Ltd., San Diego, CA, USA). Then, the hearts were removed and immediately placed in isolation buffer (STE: Sucrose 250 mM, Tris 5 mM, EGTA 1 mM; pH 7.4; 4 °C) constantly kept ice cold.

The heart was cleaned and finely chopped. Heart fragments were suspended in 10 mL of STE and homogenized using Ultra-Turrax homogenizer (IKA, T-18 Basic, IKA-Werke GmbH & Co., Staufen, Germany). The suspension obtained was subjected to centrifugations to isolate the mitochondrial component, taking care to keep it refrigerated during each step to preserve mitochondrial integrity and functionality [7,24]. Mitochondrial proteins were determined using Bradford assay.

#### 2.2.3. Cardiac Mitochondrial Membrane Potential

The membrane potential (ΔΨ) of the isolated mitochondria was determined by a potentiometric method using the liposoluble cation tetraphenylphosphonium (TPP^+^) and a selective electrode coupled with a reference one (WPI-World Precision Instruments, Sarasota, FL, USA), connected to data acquisition software (Biopac Systems Inc., Goleta, CA, USA). The electrodes were calibrated with known concentrations of TPP^+^Cl^−^ before each experiment.

The isolated mitochondria (1 mg of protein/mL) were kept on suspension with gentle and constant shaking in the incubation medium (IM: KCl 120 mM, K_2_HPO_4_ 5 mM, Hepes 10 mM, succinic acid 10 mM, MgCl_2_ 2 mM, EGTA 1 mM; pH 7.4), with the addition of TPP^+^Cl^−^ (30 μM). Grapefruit IntegroPectin was tested at concentration levels between 0.01 and 0.3 mg/mL and naringenin between 1 and 30 µM. The effects of the corresponding vehicle (DMSO 0.1%) were also evaluated. ΔΨ was calculated using the following fitted Nernst equation, as previously described [7,21]:(1)$$\Delta \psi =60log\frac{V_{0}\frac{{\left[TPP^{+}\right]}_{0}}{{\left[TPP^{+}\right]}_{t}}-V_{t}-K_{0}P}{V_{m}P+K_{i}P}$$
Δψ was expressed as millivolt (mV) changes from baseline levels. Mitochondria that showed a starting value > −170 mV were discarded as being poorly energized. All results were obtained from 6 different animals. Statistical analyses were performed with the Student’s *t*-test using the GraphPad Prism 8.0 program (Graphpad Holdings, LLC, San Diego, CA, USA), with *p* < 0.05 considered as an indicator of significant difference.

#### 2.2.4. Mitochondrial Changes in Calcium-Uptake

Mitochondrial calcium uptake was measured with a Ca^2+^ sensitive mini electrode (TIPCA, WPI, Sarasota, FL, USA) coupled to a reference one (WPI, Sarasota, FL, USA), using data acquisition software (Biopac Systems Inc., Goleta, CA, USA). Calibration curves were generated before each experiment using known concentrations of CaCl_2_. The electrodes were placed in IM with the addition of CaCl_2_ solution (100 μM) and grapefruit IntegroPectin at the concentration levels of 0.03, 0.1 and 0.3 mg/mL or naringenin 30 and 100 μM or vehicle (1% DMSO). Then, mitochondria (1 mg protein/mL) were added under gentle stirring and the changes in calcium uptake were evaluated by recording mV variations. 

Decreasing mV levels in the medium concentration of calcium was linked to its accumulation in the mitochondrial matrix. Each result was obtained with mitochondria isolated from the heart of 6 different animals. All data were expressed as mean ± SEM and were analyzed by GraphPad Prism 8.0. The data were statistically analyzed using the Student’s t-test, with *p* < 0.05 considered as an indicator of significant difference.

## 3. Results 

### 3.1. Cardioprotective Effects of Grapefruit IntegroPectin in Ex Vivo Model of Cardiac Ischemia/Reperfusion Injury

An ischemic/reperfusion episode produced marked damage to the isolated hearts of vehicle-treated rats. In this regard, a decrease in the functional parameters of myocardial contractile function (RPP) and myocardial performance (dP/dt) was observed during the reperfusion time. At the end, the levels of RPP and dP/dt were 14 ± 5% and 21 ± 9%, respectively, of the levels observed in the pre-ischemic period, as shown in Figure 2a,b. Figure 2c shows that, consistent with the functional status, the morphometric parameter also revealed marked damage and indeed the size of the ischemic area measured using formazan salt was equal to 36 ± 3%, compared to the left ventricle area.

Based on the concentration level of the flavonoids in grapefruit IntegroPectin [17], we decided to treat the rats with three different doses of grapefruit IntegroPectin (45, 135 and 450 mg/kg), in order to ensure an amount of naringin (representing 98% of total flavonoids) equal to 3, 10 and 30 mg/kg, respectively. Interestingly, in these experimental conditions, grapefruit IntegroPectin achieved dose-dependent cardioprotection, demonstrating, at the end of the reperfusion period, a significant effect with the dose of 135 mg/kg (corresponding to 10 mg/kg of naringin). In particular, the hearts derived from the rats with grapefruit IntegroPectin 135 mg/kg treatment showed, at the 120th minute of reperfusion, the RPP level of 34 ± 10%, the dP/dt level of 42 ± 9% and Ai/ALV level of 24 ± 3%, meant as percentages of the respective levels observed in the pre-ischemic period, as shown in Figure 2a–c. 

Hearts of the animals treated with naringenin showed, at the 120th minute of reperfusion, an RPP level of 42 ± 9%, dP/dt level of 38 ± 15% and Ai/ALV level of 22 ± 2%, again meant as percentages of the respective levels observed in the pre-ischemic period, as shown in Figure 2a–c. 

### 3.2. Mitochondriotropic Effects of Grapefruit IntegroPectin on Cardiac-Isolated Mitochondria

The addition of grapefruit IntegroPectin in isolated cardiac mitochondria, at the concentration levels of 0.01, 0.03, 0.1 and 0.3 mg/mL, corresponding to about 1, 3, 10 and 30 micromolar of naringin (the glycoside-derivative of naringenin), respectively, showed concentration-dependent depolarization, similar to the *Citrus* flavonoid naringenin, as shown in Figure 3a,b. Consistently, Figure 4 shows that the addition of grapefruit IntegroPectin reduced the uptake of calcium into the mitochondrial matrix in a concentration-dependent manner, suggesting that the activation of mitoK channels might be responsible for the cardioprotection shown by the extract.

## 4. Discussion

Naringin and its aglycone naringenin have shown several beneficial health effects, including cardiovascular, metabolic and, recently, anti-SARS-CoV-2 effects [25], but their absorption, distribution, metabolism and excretion, particularly in the presence of a food matrix, impact their bioavailability, which in turn affects the bioactivities of these flavonoids in vivo [26]. As mentioned in Section 1, often the encouraging in vitro results are not confirmed in in vivo approaches and clinical trials are inconsistent. The limited bioavailability of *Citrus* flavonoids can be, at least in part, the reason for these discrepancies. Indeed, *Citrus* flavonoids show poor water solubility, their bioavailability is low and they undergo extensive metabolism [26]. These pharmacokinetic complications limit their translational value and the possibility to use them as nutraceuticals in humans. In order to improve this aspect, great efforts are underway. One of the most promising paths is the use of delivery systems, such as nanotechnologies, that may allow controlled release and also targeting to specific organs or tissues. Concerning naringenin and its glycosidic form naringin, the most used nanocarriers are lipid-based, polymeric and nanoemulsions. These particles can encapsulate naringenin or naringin inside their structures and release it in vivo [27]. Another relevant aspect in the *Citrus* fruits industry is represented by the amount of waste [28]. Indeed, mesocarp and endocarp are by-products, although containing high levels of bioactive constituents, especially *Citrus* flavonoids, such as naringin [29]; therefore, there is another field of research that addresses the recovery of waste products. 

In this context, HC may be considered as an innovative green technique that allows the convenient exploitation of waste streams from the *Citrus* fruits industry, complying with the principles of bioeconomy [16]. This technique affords end products (*Citrus* fruits IntegroPectins) with higher water solubility, in which a great amount of bioactive compounds are adsorbed at the surface of pectin. Indeed, high levels of naringin (about 73 mg/g of extract) and other characteristic flavanones, as well as volatile compounds (such as limonene and alpha-linalool), have emerged from the phytochemical analyses [16,30]. 

Other commercial *Citrus* pectins have been characterized from the phytochemical and pharmacological point of view; however, they have not been found to contain relevant levels of polyphenols and their interest is mainly linked to the polysaccharide portion [19,31]. Conversely, recent studies have demonstrated that cavitation-derived IntegroPectins have more pronounced efficacy if compared to the commercial products [20]. Indeed, this HC-derived pectin not only contain plentiful flavonoids and terpenes, but it also features a unique molecular structure with very low degree of esterification and low degradation of the highly bioactive “hairy” RG-I chains [20]. This ensures immediate dissolution of the low-methoxyl grapefruit IntegroPectin in water at room temperature, whereas commercial citrus pectins (a high-methoxyl pectin virtually devoid of RG-I regions) require prolonged heating. 

In order to explore this new product from a pharmacodynamic and pharmacokinetic point of view, we carried out an in vivo treatment through intraperitoneal injection and, 2 h later, the heart was explanted and set up on Langendorff apparatus to proceed with the ischemia/reperfusion protocol. Through this type of administration, bioactive compounds do not need to be adsorbed at the intestinal level; however, they must pass in the blood and be distributed among the organs and tissues, in order to carry out pharmacological processes; among these organs, the liver can be recognized as the main district responsible for the metabolism of flavonoids. Therefore, we can discuss, even if not exhaustively, the putative improvement of bioavailability. Moreover, we chose to evaluate the anti-ischemic cardioprotection that has been previously demonstrated for naringin and naringenin [7,12,32,33,34,35,36]. The cardioprotection efficacy shown by grapefruit IntegroPectin was almost superimposable to the efficacy of the naringenin administered at the higher dose of 100 mg/kg that, according to previous studies [7], demonstrated protective properties against ischemia/reperfusion injury. In particular, grapefruit IntegroPectin showed an efficacy comparable to the purified naringenin, but with a 10 times lower dose. Moreover, it was reported in the literature that naringin also possesses effective cardioprotection against several types of myocardial insults, yet at a higher dose (in the range between 20 and 100 mg/kg) compared to that found in grapefruit IntegroPectin [32,33,34,35,36].

A deeper insight into the mechanism of action in cardiac-isolated mitochondria confirmed that the flavonoid portion, most likely naringenin/naringin, is the main actor of this promising pharmacological profile, being involved in the activation of mitoK channels. Indeed, usually an activator of mitoK channels is able to promote mild depolarization of mitochondrial membrane potential to contain the accumulation of calcium ions in the matrix, thus reducing apoptosis and preserving the cell viability. Indeed, grapefruit IntegroPectin was shown to be able to induce moderate depolarization, in addition to a reduction in intramitochondrial calcium uptake.

## 5. Conclusions

In conclusion, the results of this in vivo investigation demonstrate that grapefruit IntegroPectin displays anti-ischemic cardioprotective activity that exceeds that of the pure bioactive flavanone naringenin on a dose-dependent basis. Investigation of the mechanism of action in cardiac-isolated mitochondria confirms that the flavonoid portion, most likely naringenin/naringin, is the main actor of this promising pharmacological profile, being involved in the activation of mitoK channels. We hypothesize that the enhanced anti-ischemic cardioprotective activity of this whole *Citrus* pectin may be due to the more favorable pharmacokinetics of naringin, at a systemic level, vehiculated by the highly soluble IntegroPectin acting as a drug carrier. Further studies will be aimed at verifying this hypothesis.

This study further confirms that the emerging extraction technique based on HC processes offers a valuable opportunity for the convenient exploitation of waste *Citrus* fruits, in particular from grapefruit, with the end product presenting high concentration levels of *Citrus* flavonoids and improved bioaccessibility that enhances its nutraceutical and, consequently, economic value. 

## Figures and Tables

**Figure 1 foods-11-02799-f001:**
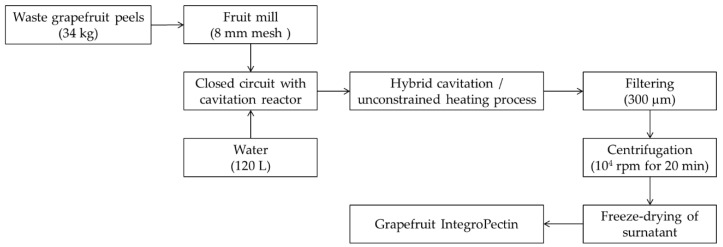
Schematic diagram for the production of grapefruit IntegroPectin.

**Figure 2 foods-11-02799-f002:**
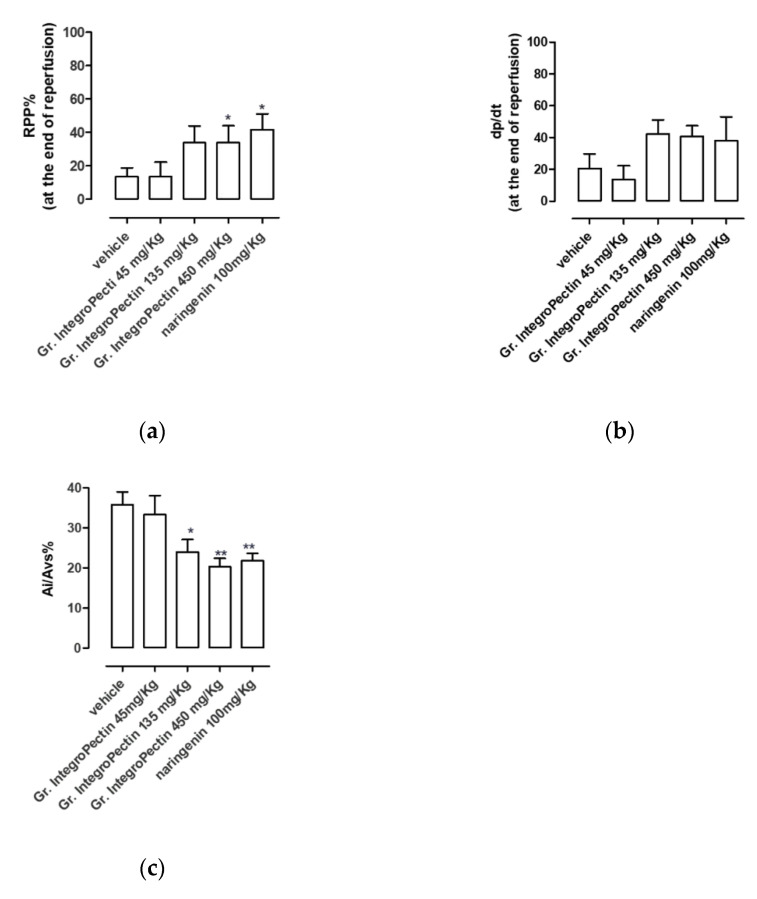
The histograms represent the functional and morphological changes induced by treatment of the animals with grapefruit IntegroPectin, naringenin or with the vehicle before the ischemia/reperfusion episode: (**a**) changes in RPP% at the end of reperfusion; (**b**) changes in dP/dt% at the end of reperfusion; (**c**) changes in the percentage of ischemic area vs. left ventricle area (AI/ALV%). The vertical bars symbolize the standard errors (*n* = 6). Asterisks show a statistically significant difference from the value observed in the hearts of vehicle-treated animals (* *p* < 0.05; ** *p* < 0.01).

**Figure 3 foods-11-02799-f003:**
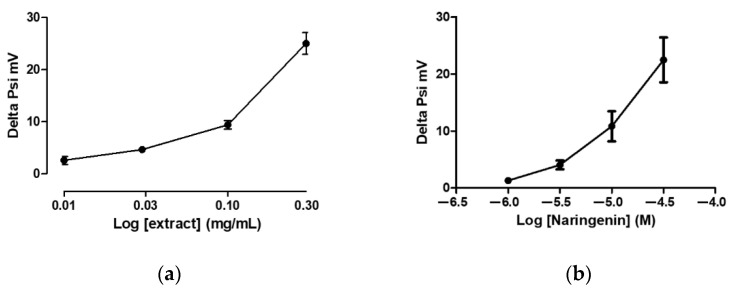
Changes in Delta Psi value (mV) following the addition, into the mitochondrial suspension buffer, of cumulatively increasing concentrations of: (**a**) grapefruit IntegroPectin; (**b**) naringenin. The vertical bars symbolize the standard errors (*n* = 6).

**Figure 4 foods-11-02799-f004:**
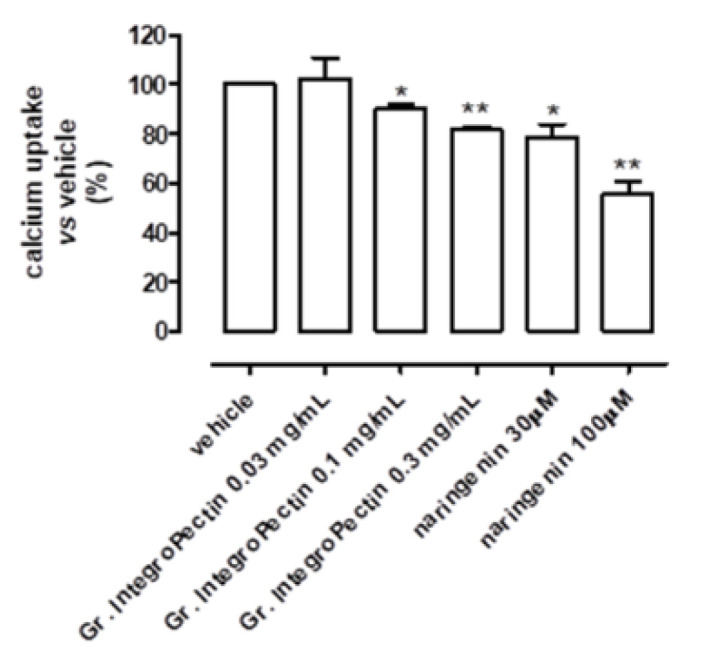
Changes in calcium uptake following the addition of increasing concentrations of grapefruit IntegroPectin or naringenin in the suspension mitochondrial buffer. The vertical bars symbolize the standard errors (*n* = 6). Asterisks show a statistically significant difference from the value observed in the hearts of vehicle-treated animals (* *p* < 0.05; ** *p* < 0.01).

## Data Availability

Data will be available from the corresponding authors upon request.

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
