# Peer review of "Cardioprotective Effects of Grapefruit IntegroPectin Extracted via Hydrodynamic Cavitation from By-Products of Citrus Fruits Industry: Role of Mitochondrial Potassium Channels"

_foods, 2022, doi:10.3390/foods11182799_

Round 1

Reviewer 1 Report

1. Why did you choose the current dose?

2. Calculate the human equivalent dose of grapefruit IntegroPectin.

3.  How the authors propose this dose grapefruit IntegroPectin should be consumed.

Reviewer 2 Report

The authors have conducted an interesting study. However, there is plenty of room for improvement. Some issues need to be resolved because the results and discussion are not well organised.

     1. Please include brief explanation on the procedure to check the purity of IntegroPectin in the Material and Methods section.

     2. Since this is an animal study, the toxicity level of Integropectin need to be done. Please include the toxicity study in the Materials and Methods section.

     3. Rewrite the Results section and only include the findings of the study. Please include the explanation of the results in the Discussion section (line 210-213) and (line 221-225).

     4. The authors need to rewrite the Discussion section because some of the information given are not discussing the findings but outside the study scope.

     5. In brief, please mention the method to check the level of naringin based on the doses of grapefruit IntergroPectin that have been used in this study (line 190-193).

Reviewer 3 Report

In this study, the authors investigated the in vivo cardioprotective effects of grapefruit IntegroPectin in a model of myocardial ischemia/reperfusion injury. This study provided interesting results and the manuscript was well written. To improve the quality of this manuscript, I recommend the following minor corrections:

In section of 2.1, the authors might provide a schematic diagram for the production of grapefruit IntegropPectin.

What is the chemical composition of IntegroPectin?

Line 95-99. Has this animal experiment been proven by Animal Care Review Committee? Please report the trial registration code.

In the discussion, line 290-293, the authors should focus on comparing the data from this study with those from previously published literature, and explained the main conclusion- “grapefruit IntegroPectin is endowed with an anti-ischemic cardioprotective activity, exceeding the pure bioactive flavanone naringenin on a dose-dependent basis”. Currently, I noticed most part of this discussion is still providing background for this study.

Round 2

Reviewer 2 Report

The manuscript has been well edited